# Just Leaf It: Accelerating Diffusion Classifiers with Hierarchical Class Pruning

## Abstract

Diffusion models, best known for high-fidelity image generation, have recently been repurposed as zero-shot classifiers by applying Bayes' theorem. This approach avoids retraining but requires evaluating every possible label for each input, making inference prohibitively expensive on large label sets. We address this bottleneck with the Hierarchical Diffusion Classifier (HDC), a training-free method that exploits semantic label hierarchies to prune irrelevant branches early and refine predictions only within promising subtrees. This coarse-to-fine strategy reduces the number of expensive denoiser evaluations, yielding substantial efficiency gains. On ImageNet-1K, HDC achieves up to 60% faster inference while preserving, and in some cases even improving, accuracy (65.16% vs. 64.90%). Beyond ImageNet, we demonstrate that HDC generalizes to datasets without predefined ontologies by constructing hierarchies with large language models. Our results show that hierarchy-aware pruning provides a tunable trade-off between speed and precision, making diffusion classifiers more practical for large-scale and open-set applications.

## 1 Introduction

Diffusion models have fundamentally reshaped the landscape of image synthesis, demonstrating an unparalleled ability to model complex data distributions conditioned on inputs like class labels or text prompts (Moser et al., 2024b; Bar-Tal et al., 2023; Frolov et al., 2024; Lugmayr et al., 2022; Ho et al., 2020). This deep, generative understanding of data unlocks capabilities that extend far beyond image creation, offering a powerful new paradigm for discriminative tasks (Goodfellow et al., 2014; Rezende & Mohamed, 2015). While traditional supervised classifiers excel in static, well-labeled scenarios, they often falter in dynamic, real-world settings. Their reliance on fixed label sets necessitates extensive retraining to accommodate new classes, and they struggle with out-of-distribution or open-set data. The rich, pre-trained representations of diffusion models, however, are uniquely suited for these challenging zero-shot, open-set, and robust classification tasks (Clark & Jaini, 2023; Chen et al., 2024b; Allgeuer et al., 2024).

Capitalizing on this potential, researchers have begun repurposing pre-trained diffusion models as diffusion classifiers (Li et al., 2023; Chen et al., 2024a). The approach is elegant in its simplicity: by leveraging Bayes' theorem, a model trained to estimate $p(\mathbf{x} \mid \mathbf{c})$ - the likelihood of image $\mathbf{x}$ given class $\mathbf{c}$ - can infer $p(\mathbf{c} \mid \mathbf{x})$ - the probability of class $\mathbf{c}$ given image $\mathbf{x}$. This allows for zero-shot inference without any label-specific retraining. The core mechanism involves evaluating the diffusion model's ability to reconstruct a noised input image under different class conditions, typically by estimating the noise prediction error.

Despite this compelling potential, a major computational bottleneck renders diffusion classifiers impractical for all but the smallest-scale problems (Ganguli et al., 2022; Clark & Jaini, 2023; Li et al., 2023; Moser et al., 2024a). Current methods must execute the expensive noise-prediction process for *every* potential class label for each input image, resulting in a computational cost that scales linearly with the size of the label set. While prior work has explored acceleration through weak pre-filtering of class labels (Li et al., 2023) or successive elimination (Clark & Jaini, 2023), these methods still treat the label space as flat and often evaluate a large majority of candidates.

To make diffusion classifiers more viable, we introduce the Hierarchical Diffusion Classifier (HDC), a novel, training-free approach that fundamentally restructures the classification process by exploiting semantic label hierarchies (see Figure 1). Instead of a flat, brute-force search, HDC employs a

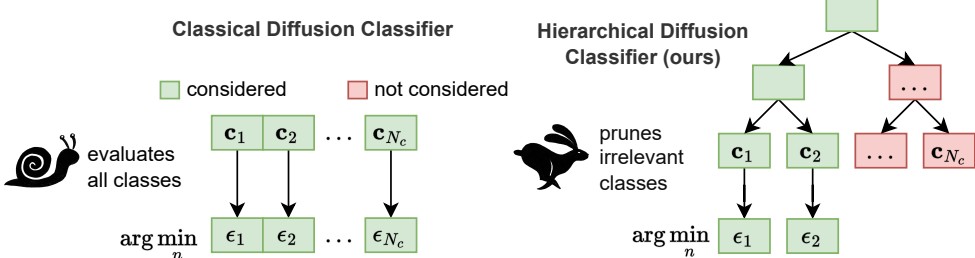

Figure 1: Comparison between the classical diffusion classifier and our proposed Hierarchical Diffusion Classifier (HDC). Whereas the classical approach evaluates all possible classes to find the correct label, which leads to unnecessary computation, HDC prunes irrelevant classes early, focusing only on the most relevant candidates. This hierarchical pruning reduces computational overhead and accelerates inference.

multi-stage, coarse-to-fine strategy. It first performs a computationally cheap evaluation at high levels of the label hierarchy (*e.g.*, "animal" vs. "vehicle"). Based on these initial scores, HDC prunes entire branches of the label tree deemed irrelevant, drastically reducing the candidate space. It then performs the standard, more computationally intensive diffusion classification *only* on the significantly narrowed set of remaining leaf-node candidates.

Our contributions are as follows:

- We propose the Hierarchical Diffusion Classifier (HDC), a training-free method that significantly accelerates diffusion-based classification by leveraging a coarse-to-fine search on a semantic label hierarchy.
- We demonstrate that HDC reduces inference time by up to 60% on ImageNet-1K while maintaining comparable accuracy, and in some configurations, even outperforming the baseline flat classifier (65.16% vs. 64.90% average per-class accuracy).
- We introduce and evaluate both fixed and adaptive pruning strategies, providing a tunable trade-off between speed and precision that enhances the feasibility of diffusion classifiers for large-scale tasks.
- We show that our approach generalizes to datasets without pre-defined hierarchies by successfully constructing and using label trees generated by Large Language Models (LLMs).

While not intended to replace standard supervised classifiers on closed-set benchmarks, HDC represents a critical step toward making diffusion classifiers practical and scalable for the dynamic, data-scarce, and open-set scenarios where their unique generative power is most needed.

## 2 RELATED WORK

Zero-shot classification enables models to recognize categories unseen during training by leveraging shared semantics between inputs and labels. CLIP exemplifies this paradigm in vision-language modeling (Radford et al., 2021), and recent large language models extend zero-shot and few-shot classification in text domains (Achiam et al., 2023; Touvron et al., 2023; Anil et al., 2023).

Diffusion models, originally developed for image synthesis (Ho et al., 2020; Dhariwal & Nichol, 2021; Rombach et al., 2022), have been adapted for discriminative use without additional training. These *diffusion classifiers* score a label by how well a conditional diffusion model reconstructs a noised input under that label (Li et al., 2023; Clark & Jaini, 2023; Chen et al., 2024b;a). This enables flexible zero-shot and open-set classification but incurs a high cost because inference scales with the number of labels: each candidate requires a forward pass (or several) through the denoiser.

To reduce this cost, prior work has introduced flat-space candidate reduction. Li et al. (2023) pre-filter labels with a weak discriminative model, while Clark & Jaini (2023) use successive elimination in a multi-armed bandit framework. These strategies lower computation yet still treat the label set as unstructured, so most comparisons remain necessary when the label space is large.

Our approach departs from flat filtering by exploiting the semantic structure among labels. We leverage dataset hierarchies (or automatically constructed label trees) to prune entire subtrees early and refine only within relevant branches. In this sense, our method complements prior accelerations while directly targeting scalability on large, structured label spaces.

# 3 PRELIMINARIES: FLAT DIFFUSION CLASSIFIER

We follow the formulation of Li et al. (2023) for extracting a zero-shot classifier from a conditional diffusion model. Let $p_\theta(\mathbf{x} \mid \mathbf{c})$ denote the likelihood of image $\mathbf{x}$ under class prompt $\mathbf{c}$. By Bayes' rule,

$$p_\theta(\mathbf{c}_i \mid \mathbf{x}) = \frac{p(\mathbf{c}_i)\, p_\theta(\mathbf{x} \mid \mathbf{c}_i)}{\sum_{j=1}^{N_C} p(\mathbf{c}_j)\, p_\theta(\mathbf{x} \mid \mathbf{c}_j)} = \frac{p_\theta(\mathbf{x} \mid \mathbf{c}_i)}{\sum_{j=1}^{N_C} p_\theta(\mathbf{x} \mid \mathbf{c}_j)}, \tag{1}$$

where we assume a uniform class prior $p(\mathbf{c}_i) = 1/N_C$.

For diffusion models trained to predict noise, the evidence lower bound links the likelihood to the $\varepsilon$-*prediction error*. Writing $\mathbf{x}_t = \sqrt{\bar{\alpha}_t}\, \mathbf{x} + \sqrt{1 - \bar{\alpha}_t}\, \varepsilon$ for the noising process with $t \in \{1, \ldots, T\}$ and $\varepsilon \sim \mathcal{N}(0, I)$, we obtain the posterior (up to normalization)

$$p_\theta(\mathbf{c}_i \mid \mathbf{x}) \propto \exp\Big\{ -\mathbb{E}_{t,\varepsilon} \left\| \varepsilon - \varepsilon_\theta(\mathbf{x}_t, \mathbf{c}_i) \right\|^2 \Big\}, \tag{2}$$

where $\varepsilon_\theta(\cdot, \mathbf{c})$ is the denoiser's noise prediction under condition $\mathbf{c}$.

**Monte Carlo estimate.** In practice, we approximate the expectation with $M$ samples $(t_k, \varepsilon_k)$:

$$\mathbb{E}_{t,\varepsilon} \left\| \varepsilon - \varepsilon_\theta(\mathbf{x}_t, \mathbf{c}) \right\|^2 \approx \frac{1}{M} \sum_{k=1}^{M} \left\| \varepsilon_k - \varepsilon_\theta\Big(\sqrt{\bar{\alpha}_{t_k}}\mathbf{x} + \sqrt{1 - \bar{\alpha}_{t_k}}\, \varepsilon_k,\, \mathbf{c}\Big) \right\|^2. \tag{3}$$

**Paired-difference (shared-sample) scoring.** For classification, only *relative* errors matter. Using the same sample set $S = \{(t_k, \varepsilon_k)\}_{k=1}^{M}$ across all classes increases statistical efficiency and yields the paired-difference approximation:

$$p_\theta(\mathbf{c}_i \mid \mathbf{x}) \approx \left[ \sum_{j=1}^{N_C} \exp\Big\{ \mathbb{E}_{t,\varepsilon}\big( \|\varepsilon - \varepsilon_\theta(\mathbf{x}_t, \mathbf{c}_i)\|^2 - \|\varepsilon - \varepsilon_\theta(\mathbf{x}_t, \mathbf{c}_j)\|^2 \big) \Big\} \right]^{-1}. \tag{4}$$

This *flat diffusion classifier* thus assigns scores to all labels and normalizes across the label set, enabling zero-shot and open-set prediction without any discriminative retraining (Li et al., 2023; Clark & Jaini, 2023; Chen et al., 2024b;a). Its main drawback is computational: inference cost scales linearly with $N_C$ because each class requires evaluating the denoiser for multiple $(t, \varepsilon)$ pairs.

# 4 HIERARCHICAL DIFFUSION CLASSIFIER (HDC)

Flat diffusion classifiers evaluate all candidate labels independently, leading to an inference cost that scales linearly with the number of classes. To alleviate this bottleneck, we introduce the *Hierarchical Diffusion Classifier* (HDC), which exploits semantic label trees to prune irrelevant branches early and restrict expensive diffusion evaluations to a small set of promising candidates.

## 4.1 TRAVERSING THE LABEL TREE

Let $T_h = (N, E)$ denote a hierarchical label tree of depth $h$, with nodes $N$ and edges $E$. Each node $n \in N$ corresponds to a synset (or a class if $n$ is a leaf). The root node is $n_{\text{root}}$, and `Children(n)` denotes its child nodes. Each node carries a label embedding $\mathbf{c}_n$. For leaves, these are class labels.

We begin with $\mathcal{S}_{\text{selected}}^1 = \{n_{\text{root}}\}$. At step $d$, for each selected node $n_s \in \mathcal{S}_{\text{selected}}^d$, we compute the $\varepsilon$-prediction error for its children:

$$\epsilon_n = \mathbb{E}_{t,\varepsilon} \left\| \varepsilon - \varepsilon_\theta(\mathbf{x}_t, \mathbf{c}_n) \right\|^2, \qquad n \in \texttt{Children}(n_s), \tag{5}$$

approximated via Monte Carlo sampling as in Equation 3, but with a smaller $M$ for efficiency.

Based on these scores, we prune nodes by retaining only those below a threshold determined by a pruning strategy (see Section 4.3). Formally,

$$\mathcal{S}_{\text{selected}}^{d+1} = \left\{ n \in \texttt{Children}(n_s) \ \mid \ n_s \in \mathcal{S}_{\text{selected}}^{d}, \ \epsilon_n \leq \text{threshold}(K_d) \right\}. \tag{6}$$

The process continues until depth $h$, where $\mathcal{S}_{\text{selected}}^{h}$ contains the final leaf candidates. The final prediction is then given by the flat diffusion classifier restricted to this pruned set:

$$\mathbf{c}_{n_{\text{final}}}, \quad n_{\text{final}} = \arg \min_{n \in \mathcal{S}_{\text{selected}}^{h}} \epsilon_n. \tag{7}$$

By pruning aggressively, HDC reduces the number of denoiser calls from $\mathcal{O}(N_C)$ to sublinear in $N_C$. If $b$ is the branching factor and $K_d$ the pruning ratio at level $d$, the cost scales as

$$\mathcal{O}\left(N_C^{1+\log_b K} M C_\varepsilon\right), \qquad 1 + \log_b K < 1, \tag{8}$$

where $C_\varepsilon$ is the cost of one $\varepsilon_\theta$ evaluation. In practice, speed-up is roughly $1/K$ compared to flat diffusion classification.

## 4.2 TREE SETUP

HDC requires a label hierarchy but is not tied to a specific source. For ImageNet-1K, we use the WordNet ontology (Deng et al., 2009), pruning overly vague nodes (*e.g.*, "entity" or "artifact") and collapsing redundant subtrees, yielding a depth-7 hierarchy. For datasets without native ontologies (*e.g.*, CIFAR-100, Food101, Oxford Pets), we construct trees using large language models to generate semantic groupings. This demonstrates HDC's flexibility: it leverages existing taxonomies when available and synthesizes plausible ones otherwise.

## 4.3 PRUNING STRATEGIES

We implement two pruning strategies:

- **Fixed Pruning.** At each level, retain the top-$K_d$ fraction of nodes with lowest error scores.
- **Dynamic Pruning.** At each level, retain nodes within $2\sigma_d$ of the minimum error, *i.e.*,

$$\mathcal{S}_{\text{selected}}^{d+1} = \{ n \in \texttt{Children}(n_s) \ \mid \ n_s \in \mathcal{S}_{\text{selected}}^{d}, \ \epsilon_n \leq \epsilon_{\min}^{d} + 2\sigma_d \}, \tag{9}$$

  where $\epsilon_{\min}^{d}$ and $\sigma_d$ denote the minimum and standard deviation of error scores at depth $d$.

Fixed pruning provides explicit control over the speed–accuracy trade-off, while dynamic pruning adapts automatically to the score distribution. Both lead to substantial runtime reductions, as shown in our experiments section.

## 4.4 DYNAMIC CLASS MODIFICATION

Unlike discriminative classifiers, HDC naturally supports dynamic class modifications. Removing a class corresponds to pruning its leaf; adding a class amounts to inserting a new leaf under an appropriate parent (either predefined or selected greedily). This property makes HDC particularly suited to open-set and evolving label spaces.

## 5 EXPERIMENTS

This section presents our experimental setup and results, evaluating different aspects of HDC: pruning strategies, prompt engineering, SD variations, and an overall evaluation of per-class accuracy on various datasets. Our code will be published upon acceptance.

Table 1: **ImageNet-1K comparison** of overall and per-class classification accuracy and inference time between the classical diffusion classifier (Li et al., 2023) and our proposed HDC (fixed and adaptive pruning) using Stable Diffusion 2.0. HDC achieves significant inference time reduction (up to 60%) while maintaining or improving accuracy. The best results are marked in bold, the second-best underlined.

| Method | Pruning | Avg. Accuracy [%] | | Time [s] | Speed-Up [%] |
| | | Overall | Per-Class | | |
| --- | --- | --- | --- | --- | --- |
| Flat Diffusion Classifier (Li et al., 2023) | – | 64.70 | 64.90 | 1600 | – |
| HDC (ours) | Fixed | **64.90** | **65.16** | 980 | 38.75 |
| HDC (ours) | Adaptive | 63.20 | 63.33 | **650** | **59.38** |

## 5.1 SETUP

HDC is based on the efficient framework established by Li et al. (2023), with added modifications tailored for hierarchical processing and pruning of candidate classes, further customized for diffusion classification on Stable Diffusion (SD) (Rombach et al., 2022). Yet, our method is adaptable, allowing seamless integration with different diffusion models and possible fine-tuning to support various hierarchical pruning strategies. To demonstrate this, we accommodate the SD versions 1.4, 2.0, and 2.1. For fixed pruning, we set $K_d = 0.5$ for all possible $d$-values. All evaluations were performed at $512 \times 512$, the resolution under which all versions of SD were originally trained. Also following Li et al. (2023), we used the $l_2$ norm to compute the $\varepsilon_t$-predictions and sampled the timesteps uniformly from $[1, 1000]$.

For Imagenet-1K (Deng et al., 2009), the class labels are converted to the form "a photo of a <*class label*>" using the template from the original work (Li et al., 2023). Inspired by Radford et al. (2021), we also experiment with prompt templates "A bad photo of a <*class label*>", "A low-resolution photo of a <*class label*>" and "itap of a <*class label*>". For CIFAR-100 (Krizhevsky, 2009), we use "a blurry photo of <*class label*>". Finally, for the Food101 (Bossard et al., 2014) and Pets (Parkhi et al., 2012) datasets, we use the template "a photo of a <*class label*>, a type of food/pet."

## 5.2 MAIN RESULTS

Table 1 highlights the results of our HDC with both pruning strategies (fixed and adaptive) compared to the classical, flat diffusion classifier (Li et al., 2023) on ImageNet-1K.

**Overall.** As observed, both pruning strategies show significant improvements in runtime compared to classical diffusion classifiers, and each is suited to different prioritizations of speed versus accuracy. Fixed pruning yields the best trade-off results on ImageNet-1K, achieving significant runtime reductions (up to 980 seconds) with a top-1 accuracy boost of 0.20 percentage points. By employing adaptive pruning (selecting candidates based on two standard deviations from the lowest error), we reduce the inference time even further to 650 seconds, though at the cost of a slight accuracy drop (*i.e.*, 1.50 percentage points). The adaptive strategy demonstrates that faster inference can be achieved with a small compromise in precision.

**Per-Class.** The baseline diffusion classifier achieves an accuracy of 64.90% with an inference time of 1600 seconds, providing a reference for both speed and precision. Using fixed pruning in HDC demonstrates new state-of-the-art accuracy for diffusion classifiers with 65.16%, while reducing the inference time by nearly 40% to 980 seconds. This indicates that HDC can not only improve classification performance but also leads to a considerable reduction in computation. Reducing processing time while maintaining similar accuracy makes fixed pruning a balanced choice for high-accuracy applications where inference speed is a priority. Similarly, HDC with adaptive pruning leverages dynamic pruning to further accelerate inference. While it records a slight drop in accuracy to 63.33%, adaptive pruning reduces inference time to 650 seconds - approximately 60% faster than the baseline. This strategy demonstrates the potential of HDC for use cases requiring faster response times, with only a marginal trade-off in classification performance.

Table 2: **CIFAR-100 with an LLM-generated label tree (SD 2.0).** We report class-wise Top-1 accuracy and runtime per class for the flat diffusion classifier versus HDC with *fixed* pruning ($K_d \in \{0.75, 0.5, 0.4\}$) and *adaptive* pruning (retain nodes with $\epsilon \leq \epsilon_{\min}^d + 2\sigma_d$). HDC consistently reduces inference cost while preserving or improving accuracy; *e.g.*, fixed pruning with $K_d = 0.4$ improves accuracy by +3.3 pp and cuts runtime by $\approx 34\%$ relative to the flat baseline. Best results are in **bold**, second-best are underlined.

| | Flat Diff. | Fixed Pruning | | | Adaptive Pruning |
| | Classifier (Li et al., 2023) | $K_d = 0.75$ | $K_d = 0.5$ | $K_d = 0.4$ | $\leq \epsilon_{\min}^d + 2\sigma_d$ |
|---|---|---|---|---|---|
| Class-Acc [%] | 68.93 | 56.79 | 60.57 | **72.23** | 65.12 |
| Runtime [s/class] | 1000 | **275** | 550 | 660 | 740 |
| Speed-Up [%] | - | **72.50** | 45.00 | 34.00 | 26.00 |

**Example.** Figure 2 illustrates how HDC progressively prunes the label tree. At each stage, error scores guide the elimination of unlikely branches, leaving only a small set of relevant leaf nodes. The final prediction is then obtained by applying the flat diffusion classifier on this reduced set, demonstrating how HDC shifts expensive computation to only the most promising candidates.

**Confusion.** The confusion matrix in Figure 3 shows within the "Animal" subtree. Misclassifications predominantly occur among biologically similar classes (*e.g.*, salamanders vs. lizards, or lizards vs. snakes), indicating that the model's errors are structured and semantically meaningful.

**Summary.** Our results highlight that HDC enables a tunable trade-off between inference speed and accuracy. Fixed pruning delivers the best balance of efficiency and precision, making it suitable for high-stakes classification, while adaptive pruning achieves the fastest runtimes with only minor accuracy loss.

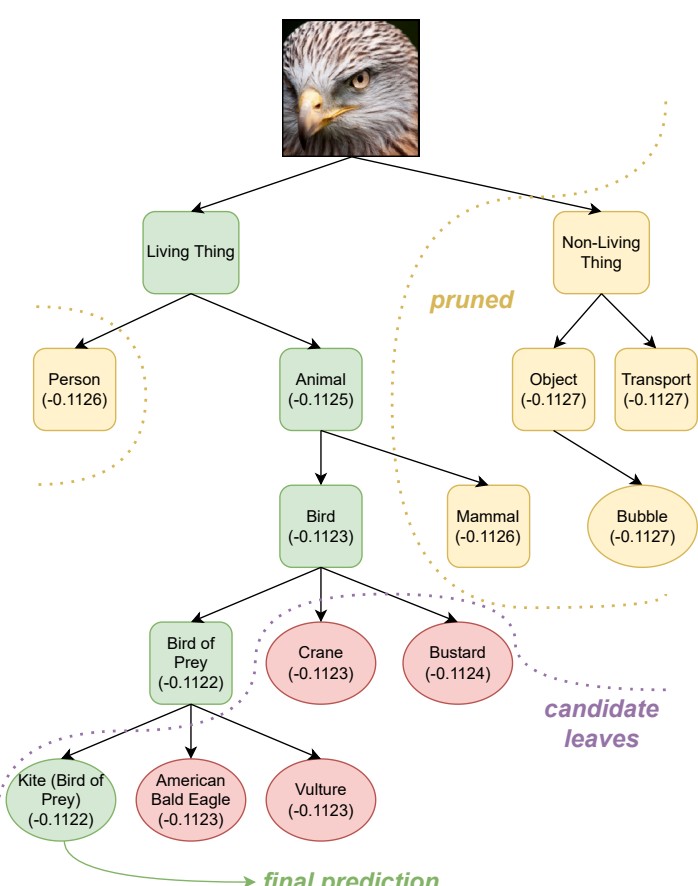

Figure 2: **Illustration of HDC on a single image.** At each stage, nodes with high error scores are pruned, leaving only relevant branches of the label tree. Here, pruning progressively narrows the candidates to semantically related classes (*e.g.*, American Bald Eagle, Vulture), before selecting the correct leaf node *Kite (Bird of Prey)* as the final prediction. This example completes in 1102 seconds, demonstrating how HDC focuses computation on a compact set of plausible labels.

## 5.3 LLM-Generated Label-Trees and Other Datasets

Our method also demonstrates notable results on CIFAR-100 when employing a LLM-generated label hierarchy, substantially outperforming the standard flat diffusion classifier baseline (see Table 2).

Table 3: Performance of HDC with fixed and adaptive pruning on **Pets** and **Food101** using LLM-generated label trees. Across both datasets, HDC accelerates inference while maintaining or improving accuracy: *e.g.*, +2.1 pp on Pets and +2.8 pp on Food101 compared to the flat diffusion classifier. Best results are shown in **bold**.

| | Pets | | | Food101 | | |
|---|---|---|---|---|---|---|
| | Top1 [%] | Top5 [%] | Time [s] | Top1 [%] | Top5 [%] | Time [s] |
| Flat Diffusion Classifier (Li et al., 2023) | 85.25 | 99.19 | 40.55 | 72.40 | **92.00** | 79.22 |
| HDC fixed (ours) | 86.53 | 98.69 | 40.00 | **75.15** | 88.95 | 66.60 |
| HDC adaptive (ours) | **87.39** | **99.39** | **40.00** | 67.00 | 81.15 | **52.60** |

Figure 3: **Confusion matrix on ImageNet-1K ("Animal" subtree).** Results shown for HDC with fixed pruning. The y-axis denotes ground-truth classes and the x-axis predicted labels (including "other classes" outside the subtree). Most confusions occur between semantically related species (*e.g.*, salamander-lizard, lizard-snake), reflecting meaningful structure in the model's errors.

Moreover, we show the influence of the pruning ratio $K_d$ for our fixed pruning strategy critically dictates the balance between classification accuracy and inference speed. For instance, decreasing $K_d$ from 0.75 to 0.5, and further to 0.4, shows a clear trend: accuracy improves from 56.79% to 60.57% and then to a peak of 72.23%, while runtime correspondingly increases from 275s to 550s and 660s.

Finally, the performance advantages of HDC generalize effectively to other datasets, such as Pets and Food101, as detailed in Table 3. For instance, on the Pets dataset, HDC adaptive improved Top-1 accuracy by +2.14pp (to 87.39%) with a negligible change in runtime (40s vs. 40.55s). On the Food dataset, HDC fixed simultaneously increased Top-1 accuracy by +2.75 percentage points (to 75.15%) and accelerated inference by a significant 16% (66.6s vs. 79.22s). Even greater speed-ups were observed with HDC adaptive on Food (52.6s, a 33.6% reduction), albeit with a trade-off in accuracy for that specific configuration.

## 5.4 STABLE DIFFUSION VERSIONS

We evaluated the HDC using different SD versions to assess its flexibility and performance across generative backbones, as summarized in Table 4. The results reveal that SD 2.0 provides the best

Table 4: Performance comparison of the HDC **with different diffusion models** using fixed and adaptive pruning for **ImageNet-1K**. Top-1 accuracy and inference time (in seconds) are reported for each SD version, highlighting SD 2.0 as achieving the highest accuracy, while adaptive pruning in SD 1.4 yields the fastest inference time.

| SD Version | Fixed Pruning | | | | Adaptive Pruning | | | |
|---|---|---|---|---|---|---|---|---|
| | Top 1 [%] (class-wise) | Top 1 [%] (overall) | Time [s] | Speed-Up [%] | Top 1 [%] (class-wise) | Top 1 [%] (overall) | Time [s] | Speed-Up [%] |
| SD 1.4 | 52.71 | 52.60 | 1000 | 37.50 | 54.77 | 54.80 | **710** | **55.63** |
| SD 2.0 | **65.16** | **64.90** | 980 | 38.75 | **63.33** | **63.20** | 980 | 38.75 |
| SD 2.1 | 61.15 | 61.00 | **950** | **40.63** | 60.91 | 60.70 | 720 | 55.00 |

Table 5: Evaluation **across different prompt types** for HDC using fixed and adaptive pruning on **ImageNet-1K**. The standard prompt, "A photo of a *<class label>*", consistently yields the highest Top-1, Top-3, and Top-5 accuracy. Alternative prompts, such as "A bad photo of a *<class label>*" and "A low-resolution photo of a *<class label>*", result in slight decreases in accuracy, showing that prompt variations can impact model performance.

| Pruning | Prompt-Type | Top 1 [%] | Top 3 [%] | Top 5 [%] |
|---|---|---|---|---|
| fixed | "A photo of a *<class label>*" | **64.90** | **80.20** | **85.30** |
| | "A bad photo of a *<class label>*" | 59.90 | 79.60 | 84.90 |
| | "itap of a *<class label>*" | 61.37 | 81.33 | 86.30 |
| | "A low-resolution photo of a *<class label>*" | 57.50 | 76.46 | 80.94 |
| adaptive | "A photo of a *<class label>*" | **63.20** | **82.30** | **86.30** |
| | "A bad photo of a *<class label>*" | 62.30 | 80.10 | 85.90 |
| | "itap of a *<class label>*" | 57.80 | 78.20 | 82.30 |
| | "A low-resolution photo of a *<class label>*" | 57.50 | 76.46 | 80.94 |

trade-off between accuracy and inference time. Specifically, when using fixed pruning, SD 2.0 achieved the highest Top-1 accuracy at 64.14% with an inference time of 980 seconds. In contrast, SD 1.4 demonstrates the fastest inference time of 710 seconds when paired with adaptive pruning, albeit with a significant top-1 class-accuracy reduction to 54.77%.

## 5.5 PROMPT ENGINEERING

Inspired by Radford et al. (2021), we also evaluated different prompt templates to assess their impact on accuracy and inference time, as shown in Table 5. The default prompt, "a photo of a *<class label>*," consistently achieved the best performance, suggesting that a straightforward prompt yields robust results across classes. Other templates, such as "a bad photo of a *<class label>*" and "a low-resolution photo of a *<class label>*," resulted in a slight drop in accuracy without significantly affecting inference time.

The rationale for testing alternative prompts stems from a hypothesis that prompts hinting at lower-quality images might help the classifier generalize better to real-world cases with variable quality, capturing diverse visual characteristics. For instance, using terms like "bad" or "low-resolution" was expected to enhance robustness to noisy or degraded inputs.

Interestingly, however, the results show that the simpler, unmodified prompt performs best, indicating that the hierarchical model likely benefits from a more neutral prompt format when dealing with high-quality image data like ImageNet-1K. Nevertheless, these prompt variations may still hold potential for datasets with inherently low-resolution or distorted images, where quality-based prompts could help the classifier learn more generalized features.

We also observed a significant disparity in inference times across specific classes, such as "snail" (221 seconds) versus "keyboard space bar" (1400 seconds). This difference likely reflects the complexity of visual features within each category: classes with intricate or ambiguous features may require longer processing times due to the hierarchical classification structure.

## 6    LIMITATIONS & FUTURE WORK

Although HDC delivers substantial speed-ups and competitive accuracy, including in robust and zero-shot open-set settings with dynamic class modifications, several limitations open avenues for future research.

Most importantly, the efficiency gains hinge on the depth and balance of the label hierarchy. Datasets with shallow trees or weak semantic groupings may see limited acceleration. This motivates the development of more sophisticated, data-driven hierarchies that can adapt to the structure of each dataset. Likewise, our method has yet to be tested on domains with highly complex or overlapping categories, such as medical imaging or fine-grained visual recognition. These scenarios present opportunities to extend HDC with adaptive thresholds, weighted traversal paths, or hybrid pruning strategies that emphasize fine-grained discriminative cues.

Looking forward, we see three especially promising directions: (i) automated hierarchy construction using LLMs or representation learning, (ii) tighter integration with multimodal diffusion models to support cross-domain classification, and (iii) dynamic, task-aware pruning strategies that adapt in real time. Together, these directions point to a broader research agenda: turning diffusion classifiers into scalable, flexible, and general-purpose tools for large-scale recognition.

## 7    CONCLUSION

We presented the Hierarchical Diffusion Classifier (HDC), a training-free framework that makes diffusion-based classification practical at scale. By replacing flat evaluation with a coarse-to-fine search over semantic label hierarchies, HDC prunes entire branches early and focuses computation only on the most promising candidates. This simple but powerful idea yields up to a 60% reduction in inference time while matching, or even surpassing, the accuracy of flat diffusion classifiers.

Beyond efficiency, HDC introduces a new design principle for diffusion classifiers: exploiting structure in the label space. Our experiments show that both fixed and adaptive pruning strategies deliver flexible control over the speed–accuracy trade-off, enabling deployment in settings ranging from high-accuracy benchmarks to real-time applications. Crucially, HDC generalizes across datasets with and without predefined hierarchies, demonstrating that scalable diffusion classification is achievable even in dynamic, open-set environments.

In short, HDC expands the role of diffusion models from generative foundations to competitive, large-scale classifiers. We believe this hierarchical perspective opens the door to a new class of diffusion-based methods that are not only expressive but also efficient, adaptive, and ready for real-world recognition tasks.

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
