# Supplementary Material for Just Leaf It: Accelerating Diffusion Classifiers with Hierarchical Class Pruning

## A  Theoretical Speed-up Analysis

In the following, we make the simplifying assumptions and definitions:

- A perfectly balanced $b$-ary tree of depth $h$, so the total number of leaf classes is $N_C = b^h$.
- A constant pruning ratio $0 < K \leq 1$ at each level.
- A fixed number of Monte-Carlo samples $M$ per evaluated node.
- Each Monte-Carlo sample requires one diffusion-model forward pass, at cost $C_\varepsilon$.

**1. Cost of the Standard (Flat) Diffusion Classifier.**  Evaluates all $N_C$ classes with $M$ Monte Carlo samples each:

$$C_{\text{flat}} = N_C \times M \times C_\varepsilon = b^h M C_\varepsilon. \tag{1}$$

**2.  Cost of the Hierarchical Diffusion Classifier (HDC).**  At level $d$ ($1 \leq d \leq h$), HDC visits $S_d = b^d K^{d-1}$ nodes. Summing over all levels:

$$C_{\text{HDC}} = \sum_{d=1}^{h} S_d \times M \times C_\varepsilon = M C_\varepsilon \sum_{d=1}^{h} b^d K^{d-1}$$

$$= M C_\varepsilon \, b \sum_{i=0}^{h-1} (bK)^i = M C_\varepsilon \, b \, \frac{(bK)^h - 1}{bK - 1}. \tag{2}$$

**3. Theoretical Speed-up Ratio**

$$S = \frac{C_{\text{flat}}}{C_{\text{HDC}}} = \frac{b^h M C_\varepsilon}{M C_\varepsilon \, b \, \dfrac{(bK)^h - 1}{bK - 1}} = \frac{b^{h-1} \, (bK - 1)}{(bK)^h - 1}. \tag{3}$$

Note that as $K \to 1$, $S \to 1$, and for $K < 1$ the speed-up increases up to roughly $1/K$.

**4. Asymptotic Complexity in $N_C$**  Since $N_C = b^h$, we have $h = \log_b N_C$. Then

$$C_{\text{flat}} = \mathscr{O}\big(N_C \cdot M \cdot C_\varepsilon\big), \quad C_{\text{HDC}} = \mathscr{O}\big((bK)^h \cdot M \cdot C_\varepsilon\big).$$

But

$$(bK)^h = (bK)^{\log_b N_C} = b^{\log_b N_C} K^{\log_b N_C} = N_C \times N_C^{\log_b K} = N_C^{1 + \log_b K}.$$

Thus, HDC scales *sublinear* in the number of classes $N_C$.

## B  Additional Method Visualization and Algorithm

Figure 1 illustrates the classification pipeline. Starting from an input image **x**, Gaussian noise $\varepsilon \sim \mathscr{N}(0, I)$ is added to generate noisy variants $\mathbf{x}_t$ across multiple diffusion timesteps $t$. A diffusion-based classifier then operates on these noisy samples using hierarchical textual prompts (e.g., "A photo of a synclass / class name"), enabling coarse-to-fine classification guided by the label tree.

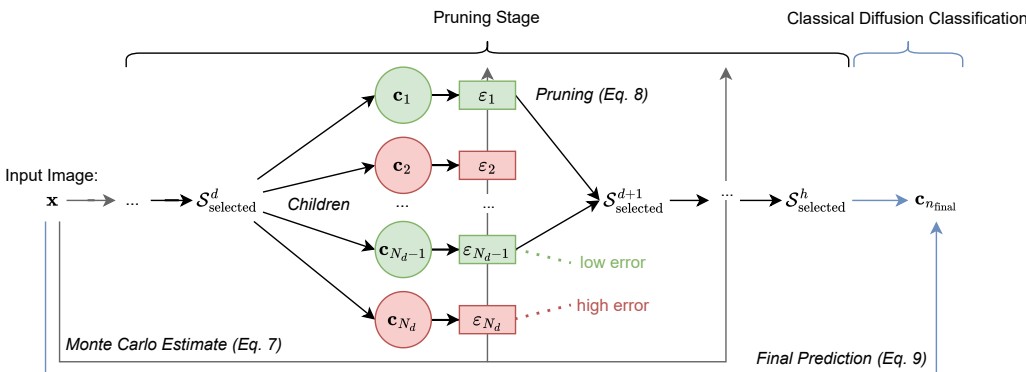

Figure 1: Overview of our Hierarchical Diffusion Classifier (HDC). Starting with an input image $\mathbf{x}$, noise $\varepsilon \sim \mathcal{N}(0, I)$ is added to generate a noisy image, resulting in $\mathbf{x}_t$ for multiple timesteps $t$. Next, we use the diffusion classifier with a reduced number of $\varepsilon$-predictions and hierarchical conditioning prompts like "A photo of a {synclass / class name}" to progressively refine the classification through multiple levels of the label tree. By doing so, we keep track of the most promising classes (highlighted in green) and ignore the rest (highlighted in red). The set of selected nodes during the pruning stage is denoted as $\mathscr{S}^d_{\text{selected}}$, where $d$ denotes the step count during traversal from 1 to $h$, the depth of the tree. Subsequently, the classical diffusion classifier pipeline is applied to the pruned, more specific subcategories (leaf nodes), which results in faster classification overall.

At each level of the hierarchy, the classifier evaluates a reduced set of candidate classes, identifying the most promising ones (shown in green) and discarding others (shown in red). The set of retained class nodes at depth $d$ is denoted by $\mathscr{S}^d_{\text{selected}}$, as the classifier traverses from the root down to depth $h$. Finally, the diffusion classifier is applied more thoroughly to these pruned leaf nodes, significantly accelerating inference by focusing only on the most relevant subcategories.

Algorithm 1 formalizes this pruning process. For each level of the label tree, it computes prediction errors across a small number of Monte Carlo samples and retains the top-$K_d$ child nodes based on these errors. This iterative pruning continues until the classifier reaches the leaf level, at which point the final subset of class labels is returned for final classification. All experiments using this algorithm were conducted on an RTXA6000 GPU with 100G memory.

## C  DATASET HIERARCHY CREATION

We organize our datasets into multilevel label trees—reusing an existing taxonomy whenever possible (for example, ImageNet's WordNet–derived hierarchy or CIFAR-100's two-tier structure). When explicit hierarchy exists, we prompt a large language model to create one by using the prompt: "Create a hierarchy for the dataset <dataset> with X levels, ending in the leaf-node class labels."

This approach works even when the semantic gap between coarse- and fine-groups is small: we exploit subtle inter-class distinctions. For instance, in the Oxford-IIIT Pets dataset we ask for a two-level taxonomy: first "Cat" vs. "Dog," then the individual breed names as leaves. Furthermore, we can ask the LLM to group classes by color palettes ("red object," "green object"), by shape ("square-shaped vs. oval-shaped"), or by any other perceptual or conceptual feature.

This highlights the domain adaptability of our method as well as its flexibility to be applied on new datasets or even open-set ones.

---

**Algorithm 1** Hierarchical Diffusion Classifier (HDC) in the pruning stage for classifying one image

---

**Input:** test image $\mathbf{x}$, $T_h = (N, E)$ with nodes $N$, edges $E$ and depth $h$, root node $n_{\text{root}}$, label inputs $\{\mathbf{c}_i\}_{i=1}^{N_c}$, pruning ratios $K_d$, and number of random samples $M$.

1: *// initialization*
2: Selected = list(Children($n_{\text{root}}$))
3: Errors = dict()
4: ErrorsCalculated = dict()
5: **for** each node $n \in N$ **do**
6:     Errors[$\mathbf{c}_n$] = list()
7:     ErrorsCalculated[$\mathbf{c}_n$] = false
8: **end for**
9:
10: *// modified diffusion classifier error calculations*
11: **for** tree depth $d = 1, \ldots, h$ **do**
12:     **for** stage $i = 1, \ldots, M$ **do**
13:         Sample $t \sim [1, 1000]$
14:         Sample $\varepsilon \sim \mathcal{N}(0, I)$
15:         $\mathbf{x}_t = \sqrt{\bar{\alpha}_t}\mathbf{x} + \sqrt{1 - \bar{\alpha}_t}\varepsilon$
16:
17:         *// calculate child errors*
18:         **for** each node $n_s$ in Selected **do**
19:             **for** each child node $n \in$ Children($n_s$) **do**
20:                 *// check if error already calculated*
21:                 **if** ErrorsCalculated[$\mathbf{c}_n$] **then**
22:                     continue
23:                 **end if**
24:
25:                 Errors[$\mathbf{c}_n$].append($\|\varepsilon - \varepsilon_\theta(\mathbf{x}_t, \mathbf{c}_n)\|^2$)
26:             **end for**
27:         **end for**
28:     **end for**
29:
30:     *// descend in the tree and select top-k*
31:     ErrorsCalculated[Selected] = true
32:     SelErrors = mean(Errors[Selected])
33:     Selected = TopK(SelErrors, $K = K_d$)
34: **end for**
35:
36: *// return pruned class label set*
37: **Return:** Selected

---