# OpenReview forum: "Just Leaf It: Accelerating Diffusion Classifiers with Hierarchical Class Pruning"
_ICLR.cc/2026/Conference — ICLR 2026 Conference Withdrawn Submission_

### Official Review · Reviewer_t5x7 · 2025-10-22

**Soundness:** 2
**Presentation:** 3
**Contribution:** 1
**Rating:** 2
**Confidence:** 4

**Summary:**

This paper proposes HDC (Hierarchical Diffusion Classifier) to improve the inference efficiency of diffusion classifiers. Unlike "flat" approaches that evaluate the diffusion model against all possible classes, HDC utilizes a class label hierarchy to iteratively prune unlikely branches of the label tree. This structured pruning mitigates the key limitation of diffusion classifiers, where computational cost scales linearly with the number of classes. The hierarchy can be generated using Large Language Models (LLMs) for datasets lacking a canonical hierarchy (e.g., Food101 or Pets). Empirical results demonstrate that HDC can achieve significant inference time reductions while incurring only a minor drop in classification accuracy.

**Strengths:**

- The proposed method is simple, intuitive, and training-free.
- The paper is well-written, and the core idea is presented clearly and is easy to follow.
- The motivation of the proposed method to accelerate diffusion classifiers is clear.

**Weaknesses:**

- **Questionable Significance of the Problem:** While improving efficiency is a valid practical goal, the paper's core premise, that diffusion classifiers are a "promising" direction for zero-shot, open-vocabulary classification (L36), is questionable. It seems the task can be already handled well by highly effective CLIP-based models [1] and, more recently, powerful Multimodal Large Language Models (MLLMs) [2]. The paper fails to provide a compelling argument for why the research community should focus on optimizing diffusion classifiers for this task when established and often more efficient alternatives already exist.
- **Missing Key Empirical Comparisons:** The paper cites [3], which also proposes a pruning method ("candidate class pruning") for diffusion classifiers, but provides no empirical comparison. Furthermore, to substantiate the claim of practical utility, a computational cost comparison against aforementioned CLIP and MLLMs if possible.
- **Insufficient Analysis of the Pruning Method:** The paper lacks in-depth analysis to elevate the method beyond a simple heuristic or implementation detail. For example, the impact of different pruning strategies, the sensitivity to the tree structure, and the justification for the specific adaptive pruning method are not thoroughly explored (see Questions 1, 3, 4).
- **Weak Justification for Hierarchy Availability:** The method's effectiveness is contingent on a high-quality class hierarchy. The paper does not adequately discuss the general availability of such hierarchies. While LLMs are proposed as a generator, the robustness, quality, and potential biases of the generated hierarchies are not analyzed in depth.
- **Questionable Practical Benefit**: In practice, evaluating the "flat" baseline across all classes can be parallelized. For the extreme case of maximum parallelization, HDC will not show any speedup. This parallelization would substantially diminish the practical wall-clock speedup of HDC, potentially rendering the theoretical gains marginal in a real-world scenario. The authors should discuss this limitation and provide runtime analysis in a batched, parallel setting.

[1] Your Diffusion Model is Secretly a Zero-Shot Classifier, ICCV2023

[2] Revisiting MLLMs: An In-Depth Analysis of Image Classification Abilities, arXiv 2412.16418

[3] Text-to-Image Diffusion Models are Zero-Shot Classifiers, NeurIPS 2023

**Questions:**

1. The paper claims a tunable speed-precision trade-off (L84). Could the authors validate this by providing a complete speed-accuracy curve, populated by densely sampling the pruning ratio? Table 2 only provides a few coarse data points and omits cases with more extreme values of pruning ratio.
2. An unexpected result in Table 2 is that HDC sometimes imporves accuracy over the non-pruned baseline. This is counter-intuitive; I believe HDC without pruning should be equivalent to the flat diffusion classifier baseline. Could the authors elaborate on this phenomenon?
3. Could the authors provide a deeper analysis of the pruning strategy itself? For instance, how does the proposed adaptive pruning compare to other possible pruning strategies? If the proposed strategy is particularly effective, what is the reason behind it?
4. The cost analysis appears to assume a perfectly balanced tree. How well do the constrcuted hierarchies adhere to this assumption? In a "worst-case" scenario with many disjoint, fine-grained classes (e.g., MNIST), HDC will provide no efficiency gain. How likely is this scenario in real-world tasks, and how does the method perform under such conditions?
5. Could the authors provide more details on generating hierarchy with LLMs? It seems ths provided information is not sufficient to reproduce experiments. Which LLM is used and how does the resulting hierarchy look like? What is the prompt used for each dataset?
6. It seems Eq. (2) is not exactly true. My understanding is that the connection between the elbo (the diffusion model's prediction error) and the posterior is purely based on an approximation (e.g., as done in [1]), not a direct proportionality.
7. The interpretation of Figure 3 is not convincing. While the 'salamander-lizard' example is supportive, the 'Other Classes' category constitutes a large portion of the misclassifications. This suggests many errors are not hierarchically "close calls" (e.g., an animal misclassified as a non-animal), which contradicts the authors' interpretation. Additionally, the presence of a 'Rest' node under 'Animal' suggests the generated hierarchy is suboptimal ("a photo of rest" is semantically ill-defined).
8. It seems some numbers in Tab.5 are bolded in a wrong way (Top-3 and Top-5 accuracies).

---

### Official Review · Reviewer_6Wzw · 2025-10-24

**Soundness:** 3
**Presentation:** 3
**Contribution:** 1
**Rating:** 2
**Confidence:** 3

**Summary:**

The paper introduces the Hierarchical Diffusion Classifier (HDC), a training-free method that accelerates zero-shot classification using pre-trained diffusion models by leveraging semantic label hierarchies to prune irrelevant branches early and focus computations on promising subtrees. This coarse-to-fine approach addresses the computational bottleneck of traditional diffusion classifiers, which evaluate every possible label, achieving up to 60% faster inference on ImageNet-1K while maintaining or slightly improving accuracy (e.g., 65.16% vs. 64.90% per-class accuracy). The method includes fixed and adaptive pruning strategies for tunable speed-accuracy trade-offs and generalizes to datasets without predefined hierarchies by constructing them with large language models, demonstrating effectiveness on CIFAR-100, Food101, and Pets.

**Strengths:**

Pros:
1. The paper introduces a training-free Hierarchical Diffusion Classifier (HDC) that leverages semantic label hierarchies for coarse-to-fine pruning, effectively addressing the computational inefficiency of flat diffusion classifiers.
2. HDC demonstrates substantial efficiency gains, achieving up to 60% faster inference on ImageNet-1K while maintaining or even surpassing baseline accuracy.

**Weaknesses:**

The theoretical and algorithmic innovations of this paper are limited. Specifically, while the paper claims to introduce a novel hierarchical pruning strategy for diffusion classifiers, the core idea of using label hierarchies for coarse-to-fine evaluation and pruning irrelevant branches closely resembles the hierarchical evaluation strategy proposed in concurrent work (Hierarchical Prompting for Diffusion Classifiers, Ning et al., ACCV 2024), which also leverages class tree taxonomies to accelerate inference in diffusion-based classification; the absence of citation or differentiation from this similar method diminishes the perceived originality.

**Questions:**

The novelty is my main concern. This paper is closely related to Hierarchical Prompting for Diffusion Classifiers, Ning et al., ACCV 2024.

---

### Official Review · Reviewer_QEf8 · 2025-10-30

**Soundness:** 2
**Presentation:** 3
**Contribution:** 3
**Rating:** 2
**Confidence:** 4

**Summary:**

The paper presents Hierarchical Diffusion Classifier (HDC), a method built upon Diffusion Classifiers. To address the inefficiency of diffusion classifiers, HDC aims to prune irrelevant classes early in the inference process. This is achieved by constructing hierarchical label trees using WordNet or large language models (LLMs). The results demonstrate that HDC not only improves efficiency but also achieves better accuracy compared to the standard diffusion classifier.

**Strengths:**

1.	The work addresses a timely and important problem. Diffusion classifiers are indeed slow, and improving their efficiency is necessary.
2.	The proposed method is simple yet effective, especially when dealing with datasets that contain a large number of classes.

**Weaknesses:**

1.	Scalability to larger diffusion models (e.g., SD3-m, FLUX) is not discussed. This is particularly crucial and relevant, as efficiency becomes even more critical.
2.	Baselines are limited. Including discriminative baselines (e.g., CLIP) or Discffusion would provide a more realistic assessment of the method’s effectiveness in practical settings.

- Discffusion: Discriminative Diffusion Models as Few-shot Vision and Language Learners (TMLR 2024)

3.	The framework may not generalize to compositional scenarios, where Diffusion Classifiers have been originally shown to perform better than in standard image classification tasks.

**Questions:**

1. It is unclear why HDC sometimes achieves better accuracy than the standard diffusion classifier. What accounts the most for the improvement?
2. Which LLM models did the authors use?
3. The experimental details are insufficient. How many timesteps and noises are tested?

4.	The paper did not discuss several related studies.:
   - A Simple and Efficient Baseline for Zero-Shot Generative Classification (arXiv, 2024)
	- Text-to-Image Diffusion Models Are Zero-Shot Classifiers (NeurIPS 2024)
	- Diffusion Classifiers Understand Compositionality, But Conditions Apply (NeurIPS 2025)

---

### Official Review · Reviewer_RF5n · 2025-10-31

**Soundness:** 3
**Presentation:** 2
**Contribution:** 3
**Rating:** 6
**Confidence:** 2

**Summary:**

The paper proposes a diffusion-based classifier that significantly improves the computation efficiency compared to the classical diffusion classifier. By leveraging a label tree, which is either given or synthesized by LLMs, the number of denoiser calls can be reduced roughly linearly to the pruning ratio of the tree. Experiments show that the proposed hierarchical diffusion classifier achieves 20~60% speed-ups compared to the classical flat diffusion classifier and also improves the accuracy.

**Strengths:**

The idea of using a label tree for hierarchical classification using diffusion models is novel and well-motivated. With the simple trick, the proposed method effectively accelerates the diffusion classifier and even improves the performance. The simplicity enables the approach to be applied to any kind of diffusion classifier and classification problems.

**Weaknesses:**

- However, the simplicity also has another side—the computational efficiency and accuracy highly depend on the quality of the class taxonomies, which is required by either human experts or LLMs whenever a new class is added to the problem.

- In addition, the amount of acceleration from hierarchical classification seems to be bounded, and the time required for classification is still extremely slower than traditional feedforward classifiers, even if the proposed approach has improved it by at most 60%. The accuracy is also far below the feedforward classifiers. This makes improving diffusion classifiers with this direction less potential.

**Questions:**

- What are the unique benefits of diffusion classifiers compared to traditional feedforward classifiers (e.g., CLIP-based zero-shot classifiers), in spite of the extremely high computational cost? Why is improving the computation efficiency of the diffusion classifier a critical problem?

- How does the quality of the label tree affect the computational efficiency as well as the accuracy of the proposed method? For example, what if we use the LLM-synthesized label tree for ImageNet-1K?

---

### Note · Authors · 2026-01-17

I have read and agree with the venue's withdrawal policy on behalf of myself and my co-authors.